# A Wavelet-Based Time-Frequency Analysis on the Supersonic Jet Noise Features with Chevrons

**Stefano Meloni** [1,*] and **Hasan Kamliya Jawahar** [2]

1   Department of Engineering, University of Roma Tre, 00146 Rome, Italy
2   Department of Aerospace Engineering, University of Bristol, Bristol BS8 1TR, UK; hasan.kj@bristol.ac.uk
*   Correspondence: stefano.meloni@uniroma3.it

**Abstract:** A detailed investigation of the statistical properties of the near-field pressure fluctuations induced by an under-expanded jet, by varying the nozzle exit shapes has been presented. Experiments using different convergent Chevron nozzles were carried out in the anechoic chamber at the University of Bristol to assess the importance of the Chevron shape on the near pressure field emitted by a single stream under-expanded jet. Measurements were carried out through an axial microphone array traversed radially to various positions for jet in an under-expanded condition at Mach number M = 1.3. The intermittent behavior is investigated considering the standard statistical indicators and a wavelet-based conditional approach, including the phase angle. The intermittent degree of various features related to different scales, such as Screech tones and broadband shock associate noise were estimated. A series of recently developed wavelet-based tools were assessed as a viable approach to investigate the noise emitted by under-expanded jets.

**Keywords:** aeroacoustic; jet noise; chevron nozzles; near-field





## 1. Introduction

One of the key aircraft noise sources that should be taken into account during the design of modern aircraft is the jet noise which dominates the take-off and the cruise phase [1]. The generation of sound by a jet exiting into a fluid medium is of great interest for several aeronautical applications and it has long been recognized by several previous studies [2,3]. A large body of literature has clarified that the dominant noise source in the subsonic case is the turbulent mixing nose, which is generated by the large-scale turbulence structures/instability waves of the jet flow [4,5]. During takeoff, when a high level of thrust is required, jet exiting flow from the engine's nozzle is under-expanded. This could also happen during the cruise phase due to the combined effect of low external static pressure and required thrust to maintain the flight Mach number in maneuvers or changing the lane [6,7]. An under-expanded jet plume is characterized by a shock cell train, which creates a series of compression and expansion into the flow, generating a further shock-associated noise [7–9]. This form of noise generated by non-ideally expanded supersonic jet comprises of two components: Screech tones and Broadband Shock Associated Noise (BBSAN). Screech tones are discrete tones that originate from an acoustic feedback loop between the shock cell train and the nozzle lip [10]. Screech is unusual among resonance phenomena, in that the resonance is entirely contained within the flow itself [11].

BBSAN, on the other hand, is generated from a weak interaction between downstream propagating large scale turbulent structures and the quasi-periodic shock cells in the jet plume. Contrary to Screech, BBSAN is present in both under and over-expanded jets. One of the characteristics of BBSAN (unlike Screech) is that the peak frequency varies as a function of observer position, a phenomenon observed in experimental data. The inclusion of these noise components in the analysis is essential to solve the problem and thus provide tolerable noise levels in the cabin to improve passenger comfort.

Aircraft manufacturers developed various technologies to reduce jet noise, and one of the most common jet technologies is the use of Chevrons at the nozzle exit [12–15]. They reduce the noise by reducing the velocity gradients in the jet shear layer and in the supersonic case they disrupt the feedback loop responsible of the Screech tone [16]. Bridges and Brown [17] showed that the increase in Chevron number and penetration achieves considerable noise reduction at lower frequencies for convergent jet.

The velocity gradient variation in the jet shear layer, induced by the Chevron presence, has been expected to modify the stochastic nature of the noise source in terms of intermitency of the aforementioned noise components. The role of intermittent events in the near-field of a jet is fundamental to evaluate the noise emitted in far-field and its role in the noise generation mechanism has been recognized in the subsonic case by several works [4,5,18,19].

In this paper, for the first time statistical analysis of the intermittency events related to the under-expanded acoustic signatures using a wavelet-based approach has been carried out. The wavelet technique is an efficient tool for extracting and analyzing the time evolution of the frequency-localized acoustic signatures. Different wavelet-based indicators, directly related to intermittency, like the local intermittency measure (*LIM*) and its square (*LIM*2), which is correlated to the flatness factor [4], was used to analyze the modification of the time evolution of the acoustic signatures detected in the baseline configurations. The time-frequency evolution of the phase-angle has been included in the analysis using a multivariate wavelet-based approach, being its modifications related to the modification of the noise directivity

To carry out this investigation, we used a database of near-field pressure signals acquired from the Bristol Jet Aeroacoustic Research Facility (BJARF) at the University of Bristol. A total of three different nozzles having different shapes of Chevrons were considered, one baseline and two chevron nozzles. The choice of nozzles were based on the availability of flow field, which is well characterized in the literature by several works (see [20,21]). According, to previous works [16,17] we consider the best nozzle configuration for noise SMC006, which has a slight penalty on the effective nozzle exhaust diameter directly correlated to the thrust and SMC002, which provided a lower noise reduction incrementing the effective nozzle exhaust diameter. The analyses were performed at M = 1.3 using a near-field microphone array positioned at h/D = 3.5 with 10 microphones covering a distance starting at the nozzle exit x/D = 0 up to x/D = 18 (D = 16.8 mm). The array position was chosen to avoid high hydrodynamic effects due to the flow grazing over the microphones. The study mainly focused on the zone close to the nozzle exit, as it is mostly dominated by Screech and BBSAN.

The paper is organized as follows: the experimental setup is reported in Section 2 and the results are illustrated in Section 3. Conclusions are presented in Section 4.

## 2. Experimental Setup

The experiments were conducted at the BJARF at the University of Bristol. The flow in the BJARF facility is conditioned and silenced using three different in-line silencers to create a quiet flow. Two silencers were placed right after the control valve outside the anechoic chamber and has a diameter of 0.3 m and a height of 1.5 m each. The third large silencer, which also acts as a plenum, was placed inside the anechoic chamber and has a diameter of 0.457 m and a height of 1.9 m. The silencers were equipped with perforated tubes for the flow with the remaining area packed with glass wool. The anechoic chamber where the tests were carried out has dimensions of 7.9 m in length, 5.0 m in width and 4.6 m in height, including the surrounding acoustic walls [16].

The tests were carried out using a round convergent nozzle (SMC 000) and two different Chevron nozzles (SMC 002 and SMC 006) with a different number of lobes, 4 and 6, respectively, as shown in Figure 1. Details about the nozzle geometries have been reported in Table 1. The Chevron nozzles were chosen from the detailed study carried out by Bridges and Brown [17].

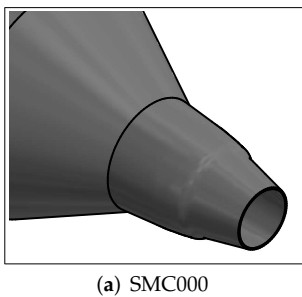

(**a**) SMC000

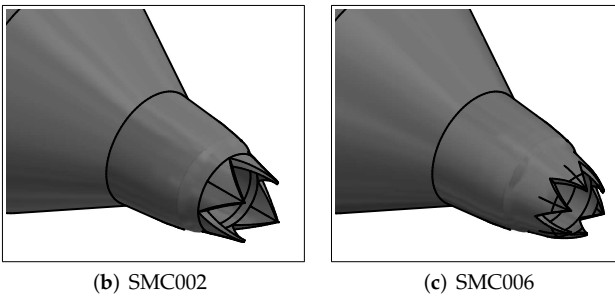

(**b**) SMC002　　　　　　　(**c**) SMC006

**Figure 1.** Schematic of the various nozzle configurations used in the present study: (**a**) SMC000 or baseline nozzle; (**b**) SMC002; and (**c**) SMC006.

**Table 1.** Parameters used for the tested chevron nozzles.

| Nozzle ID | N | Length (mm) | Angle (°) | Penetration (mm) | $D_e$ (mm) | Γ |
|---|---|---|---|---|---|---|
| SMC 000 | 0 | | | | 16.9333 | 0.089 |
| SMC 002 | 4 | 10.6667 | 5 | 0.4650 | 17.8667 | 0.089 |
| SMC 006 | 6 | 7.5333 | 18.2 | 1.1750 | 15.9000 | 0.292 |

The tested nozzles were 3:1 down-scaled version of the nozzles used by Bridges and Brown [17], which corresponds to an exit diameter of D = 16.933 mm for the SMC 000 round convergent nozzle. The tests were carried out for a wide range of supersonic flows ranging from M = 1.1 up to 1.3. Near-field unsteady pressure measurements were carried out using 1/4-inch G.R.A.S 40PL microphone that has a corrected flat frequency response at frequencies from 100 Hz to 20 kHz with a dynamic range of 150 dB. The near-filed measurements were carried with a linear array of 10 microphones placed at a distance of $2D$ away from each other, moved to 12 different heights ($h$) radially away from the jet centerline using a traverse. The closest height was chosen to be $h = 3.5D$ to avoid grazing of flow on the microphone. The data were acquired using a National Instrument PXle-4499 for $t = 12$ s at a sampling frequency of $f = 2^{17}$ Hz. To clearly show the experiment, a photo of the experimental setup is reported in Figure 2.

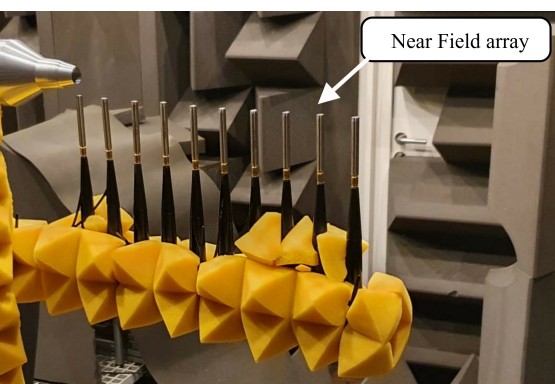

Near Field array

**Figure 2.** A photo of the experimental setup with the baseline nozzle configuration.

## 3. Results and Discussions

### 3.1. Near-Field Spectral Levels

The results for the Sound Pressure Level (SPL) for several streamwise locations ranging from x/D = 0–18 at radial location h/D = 3.5 for M = 1.3 are presented in Figure 3. The SPL was evaluated using the following equation:

$$SPL = 10\log_{10}\left(\frac{\text{PSD}\Delta f_{\text{ref}}}{P_{\text{ref}}^2}\right),\tag{1}$$

where PSD denotes the power spectral density evaluated using Welch's method, $\Delta f_{\text{ref}}$ is the frequency bandwidth and $P_{\text{ref}}$ is the reference pressure in air (equal to 20 µPa).

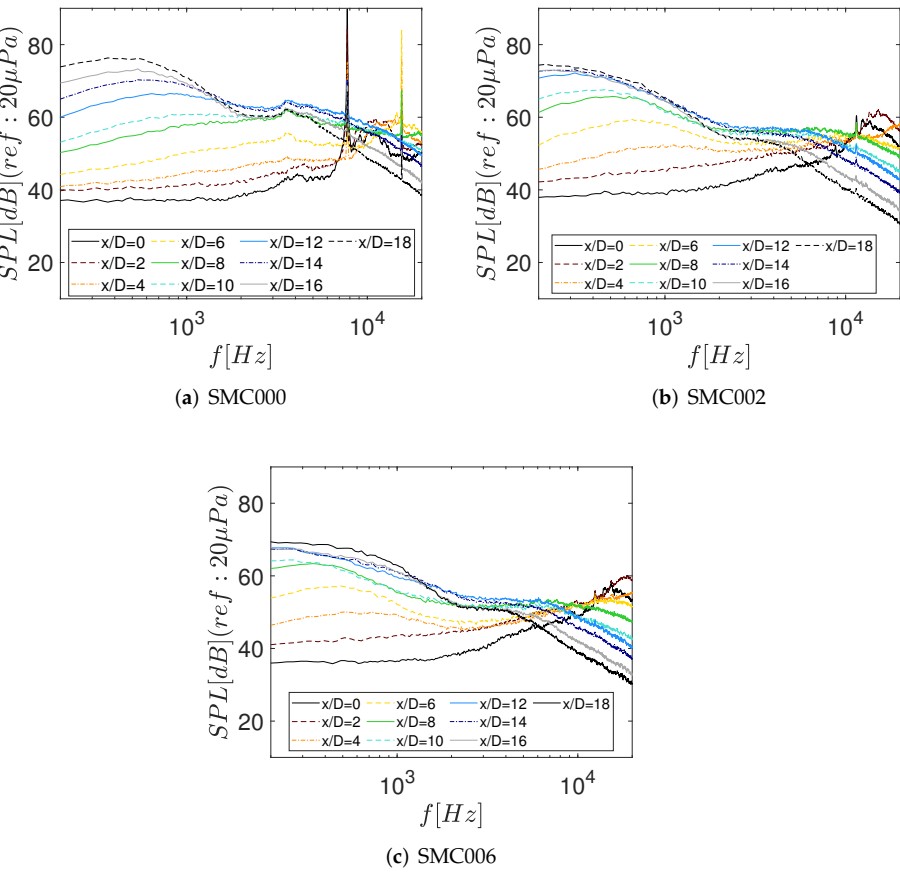

(**a**) SMC000

(**b**) SMC002

(**c**) SMC006

**Figure 3.** Near-field SPL spectra at M = 1.3 h/D = 3.5 and various axial locations. (**a**) SMC000 nozzle; (**b**) SMC002 nozzle; (**c**) SMC006 nozzle.

The results for the baseline configuration SMC000 clearly show the presence of Screech tone, after which the BBSAN spectral hump is accurately captured in Figure 3a. The results for the Chevron configurations in Figure 3b,c do not show the presence of Screech, however the BBSAN hump is present. According to the literature, [16,17] and as reported in Figure 3 the Chevrons effectively eliminate the Screech tone, which is known to be generated due to the feedback loop between the shock cells and the nozzle lip. The bump related to BBSAN results shifted to the higher frequencies for the Chevron configuration, which might be most likely due to the variation of the shear layer velocity gradient. For both the Baseline and Chevron configurations, the locations closer to the nozzle, the spectra show a flatter trend, however, at downstream locations, the spectra show a low-frequency spectral hump due to the large scale structures that are often found at the mixing region. Overall, the results show the capability of Chevron to eliminate the feedback loop, thus the Screech tone. Moreover, the results show that the number of Chevron lobes also plays

a significant role in the noise reduction mechanism as the SMC 006 with six lobes clearly shows improved noise reduction compared to SMC 002 with just four lobes.

### 3.2. Global Intermittency Analysis

One of the main aims of the present work is to provide a qualitative description of the time evolution of the near-field pressure signatures modified by the presence of the Chevrons. To have a global picture of intermittency, the third and four order statistical moments, named skewness (*s*) and kurtosis (*k*), were taken into account, which are defined in the following equations:

$$s = \frac{E[p - \mu]^3}{\sigma_p^3}, \tag{2}$$

$$k = \frac{E[p - \mu]^4}{\sigma_p^4}, \tag{3}$$

where $\mu$ is the mean of the signal $p$ and E[ ] is the expected value. As shown in Figure 4a, the evolution of the skewness factor in the stream-wise direction varies consistently between the baseline and Chevron configurations. Both negative and positive skewness can be observed for the baseline nozzle in the jet potential core region, dominated by positive pressure events within the potential core region. Predominantly, the presence of Chevrons moves the skewness to zero in the jet potential core region, and this is ascribed to the disruption of the Screech feedback loop. At the region downstream of the potential core, the near-field pressure signals have negative skewness, which is often observed in the ideally expanded case (subsonic in our case) where the Screech tone does not appear [22,23]. Negative values of the third-order statistical moments were identified in the fully turbulent jet zone at the downstream locations, which can be attributed to the development of a fully turbulent jet flow. The amplitude of these negative values increases with the Chevron presence and seems amplified by the higher number of lobes. This could be related to the increase in the velocity gradient in the jet shear layer. However, further studies about the turbulent characteristics of the nozzle are needed to clarify this assumption.

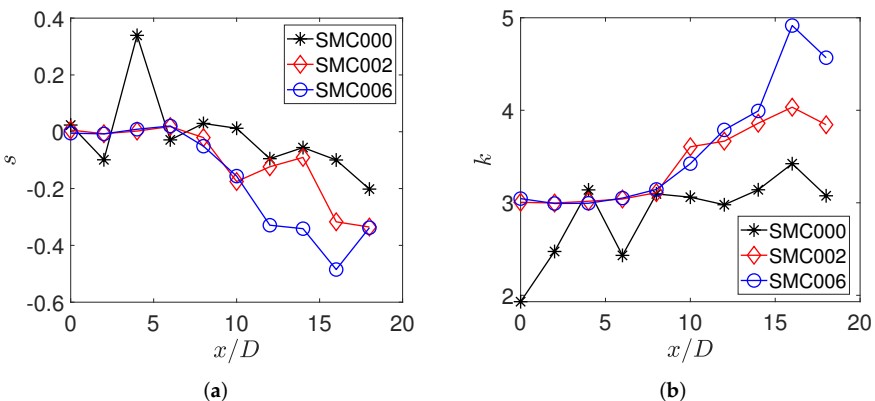

**Figure 4.** Axial evolution at M = 1.3 and h/D = 3.5 of: (**a**) Skewness; (**b**) Kurtosis.

The kurtosis trend has been reported in Figure 4b for different axial locations and different nozzle configurations. The kurtosis values were identified to be lower than three solely in the baseline configuration (SMC000) for axial positions of the microphones within the jet potential core. This could be related to the Screech presence, which is a probably persistent tonal component, further details on this will be provided with the wavelet analysis. It is important to note that the kurtosis close to 3 and the skewness close to zero indicate that the presence of Chevrons generates a Gaussian distribution of the fluctuating pressure events, suggesting a low presence of the hydrodynamic contribution.

At locations farther downstream of the nozzle exit higher kurtosis is observed, especially in the fully turbulent jet zone increasing with the number of lobes. These kurtosis

values suggest that Chevron nozzles disrupt the feedback loop related to the Screech in the first few axial locations increasing the jet flow development.

To further improve our understanding on the nature of the time signature, an analysis of the signal stochastic behavior has been carried out on a few representative cases by using the probability density functions (PDF) reported in Figure 5a,b, considering all the presented nozzle exit configuration and two different locations of the microphones in the axial direction. The pressure variable is expressed in reduced form, i.e., normalized to have zero mean value and unitary standard deviation [22].

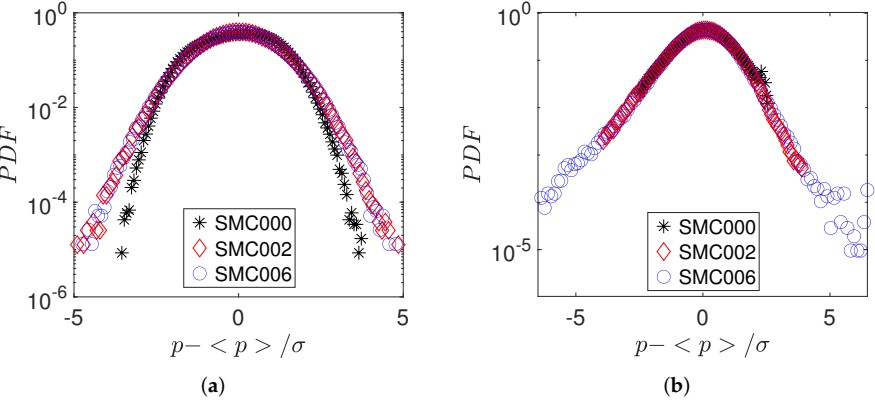

**Figure 5.** PDF comparison at M = 1.3 and h/D = 3.5: (**a**) At x/D = 2; (**b**) At x/D = 18.

A quasi-bimodal PDF has been observed in Figure 5a for the SMC000 configuration, which is due to the presence of the Screech tone. As expected, PDF shape is modified by the use of Chevron nozzles. These results also follow a Gaussian form, but with slightly larger PDF tails, which could be related to the disappearance of the Screech tone and to the quite flow developing due to the higher velocity gradient in the jet shear layer. To confirm this, at downstream location, Figure 5b, we observed a large number of pressure events in the PDF tails, especially for the SMC006 configuration, which is characterized by a higher number of Chevron lobes.

### 3.3. Single-Point Wavelet Analysis

The wavelet transform is a very proficient tool when it comes to analyzing intermittent or time-dependent features. The wavelet transform of the signal p(t) is obtained by the following expression [24,25]:

$$w(s,t) = C_\psi^{-\frac{1}{2}} s^{-\frac{1}{2}} \int_{-\infty}^{\infty} p(\tau)\psi^*\psi^*\left(\frac{t-\tau}{s}\right), \tag{4}$$

where $s$ is the wavelet scale, $\tau$ is a time shift, $C_\psi^{-\frac{1}{2}}$ is a constant that takes into account the mean value of $\psi(t)$ and $\psi^*\left(\frac{t-\tau}{s}\right)$ is the complex conjugate of the dilated and translated mother wavelet $\psi(t)$. In this analysis, we have applied the continuous wavelet transform (CWT) using the Morlet mother wavelet. The wavelet scalograms for two streamwise locations for all the three tested cases are presented in Figure 6. The first location was chosen to be within the jet potential core at x/D = 2 (see left column in Figure 6) and the second location was chosen in the turbulence mixing region at x/D = 18 (see the right column in Figure 6).

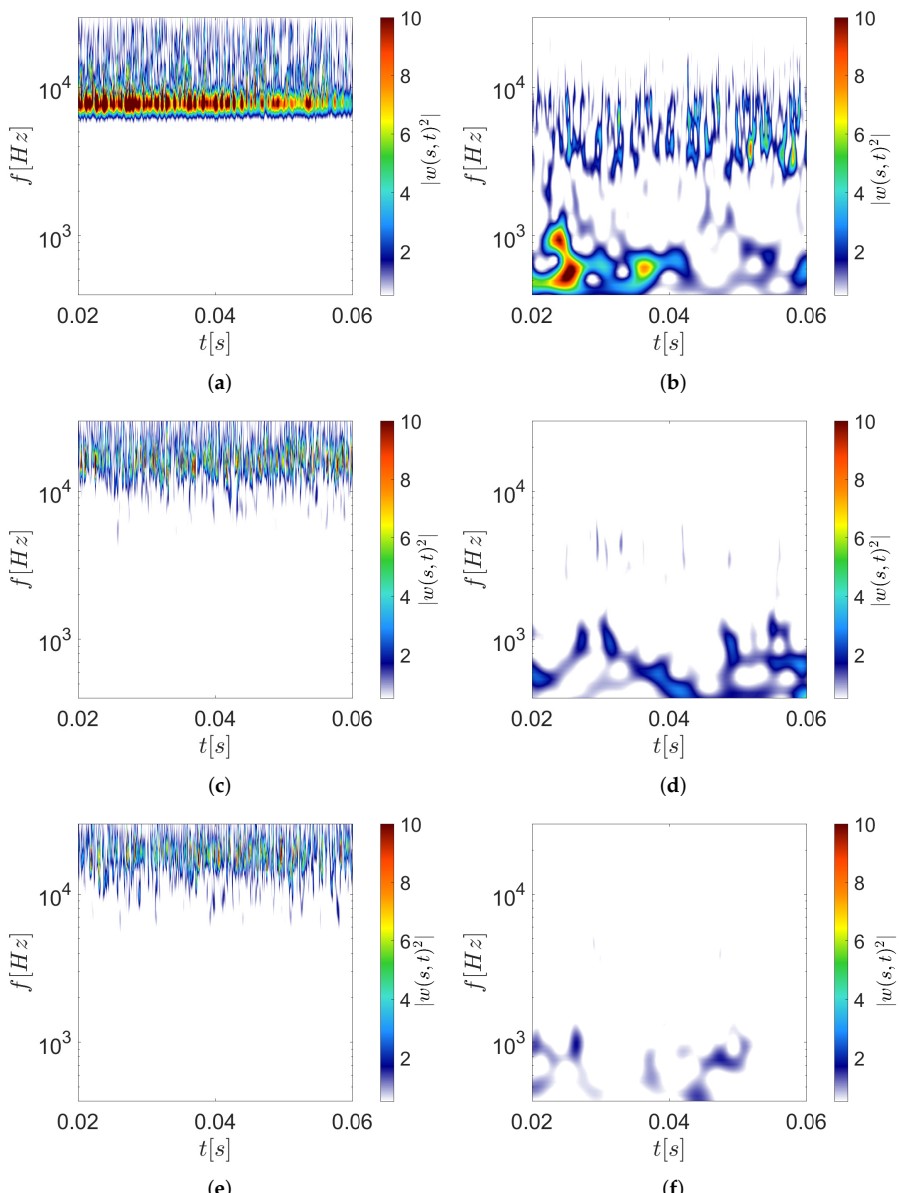

**Figure 6.** Wavelet scalograms at M = 1.3, h/D = 3.5: (**a**,**b**) Baseline nozzle at x/D = 2 and x/D = 18, respectively; (**c**,**d**) SMC002 nozzle at x/D = 2 and x/D = 18 respectively; (**e**,**f**) SMC006 nozzle at x/D = 2 and x/D = 18 respectively.

The wavelet scalogram results for the baseline configuration shows a time-persistent high energetic signature for a range of frequencies between 7 kHz and 10 kHz at x/D = 2 in Figure 6a. According to the spectral analysis seen in the previous sections, this feature is related to the Screech tone. At a higher frequency range, the same microphone location within the potential core is dominated by intermittent signatures related to the BSSAN. Moving to the characteristics of the downstream microphone at x/D = 18, Figure 6b, the intensity of the Screech signature has reduced, and energetic events intermittent in time appear at the lower frequencies because of the jet development and larger flow structures at the downstream location. It is interesting to note that although the Screech tone was observed in the spectral levels in Figure 3a at downstream locations, wavelet scalogram results show a substantial change in its intermittent characteristics compared to the upstream location. The scalogram results for the Chevron configurations, at location x/D = 2 in Figure 6c,e, show the absence of Screech signature and seems to increase the

energy of the intermittent features associated with the BBSAN. According to the spectra presented in Figure 3b,c the presence of Chevrons seems to disrupt the high energetic time signatures detected in Figure 6b at higher axial locations where the jet is developed (i.e., x/D = 18) (see Figure 6d,f). This could be the cause of the reduction of the jet mixing noise and seems amplified by the increase of the Chevron lobes.

To further understand the nature of the Screech tone and the multiple tone generation mechanism seen in the current study, the power spectral density of the time signal and the absolute values of the wavelet coefficients were calculated. The time evolution of the Screech tone frequencies at the two different axial locations were selected using the results presented in Figure 3a, considering the Screech tone and its first harmonic for the baseline configuration within the potential core region and downstream location, respectively, x/D = 2 and x/D = 18.

The results presented in Figure 7 essentially show the modulation frequency of the selected tone. The results for the first fundamental tone within the potential core shows strong frequency modulation at f = 7.7 kHz for x/D = 2. However, at x/D = 18, the spectral energy of the modulation is absent.

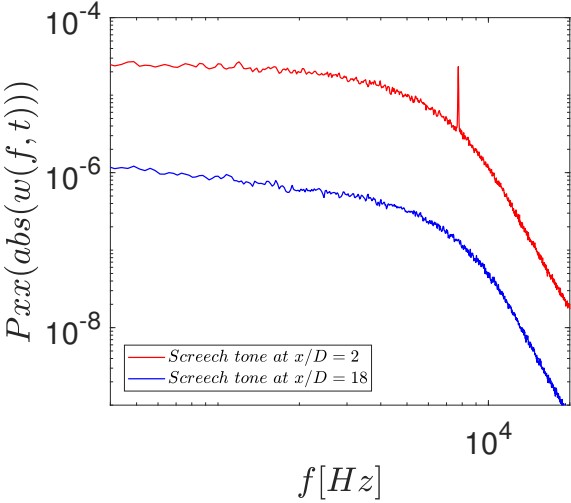

**Figure 7.** Fourier spectra of the wavelet coefficients absolute values related to the Screech tone in the baseline configuration. Results are at the same Mach number of the previous plots.

In order to present a better comparison of the fluctuating characteristics of the unsteady near-field pressure components, the statistical representation of the wavelet coefficient moduli is presented in terms of the arithmetic mean and standard deviation, according to [26] have been evaluated as follows:

$$\mu(|w_x|) = \frac{\Sigma_{i=1}^{N}|w_x|}{N}, \tag{5}$$

$$\sigma(|w_x|) = \sqrt{\frac{\Sigma_{i=1}^{N}(|w_x|_i - \mu(|w_x|))^2}{N}}, \tag{6}$$

The mean $\mu$ and standard deviation $\sigma$ of the wavelet coefficient moduli over time for each frequency were calculated using Equations (5) and (6). The results for the same for the three nozzle configurations for axial location x/D = 2 and 5 is shown in Figure 8.

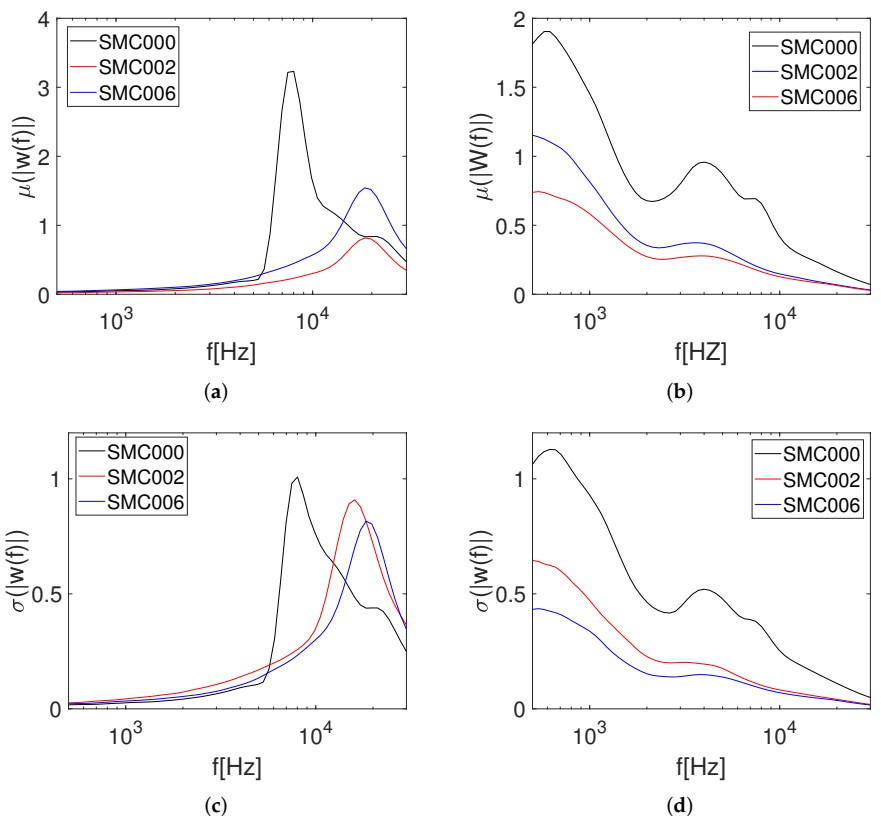

**Figure 8.** Stochastic analysis for the various frequencies of interest. (**a**) $\mu(|w_x|)$ at x/D = 2; (**b**) $\mu(|w_x|)$ at x/D = 2; (**c**) $\sigma(|w_x|)$ at x/D = 2; (**d**) $\sigma(|w_x|)$ at x/D = 18.

At first glance, Figure 8a at location x/D = 2 shows a peak in the Screech related frequency with the nozzle in the baseline configuration, which confirms the high levels of $\mu(|w_x|)$ magnitude of the Screech tone that is persistent over time. A similar trend is followed for the Screech tone for the $\sigma(|w_x|)$ results. When considering the chevron configuration, the broadband humps possess a higher magnitude compared to the baseline, with SMC006 showing high levels of magnitude at high frequencies. However, in the case of $\sigma(|w_x|)$ for the chevron configurations, the results for the SMC002 show high levels of dispersion of pressure from the mean compared to the SMC006 and SMC000. At the downstream location, x/D = 18, the $\mu(|w_x|)$ shows no signs of the Screech tone for all the three configurations; however, high levels of fluctuations could be observed at low and mid-frequency range for the baseline configuration compared to the chevrons. This could be attributed to the high levels of turbulence at the downstream location. A similar trend is followed for $\sigma(|w_x|)$, but with higher levels of dispersion of the fluctuations for the SMC002 compared to the SMC006. Overall, the results show low levels of fluctuation intensity and dispersion from the mean for the chevron configurations compared to the baseline.

To remove the dependence on the local feature energy, the so-called Local Intermittency Measure (*LIM*) [27] that represents a normalized version of the wavelet scalogram was used. Its formal definition is the following:

$$LIM(s,t) = \frac{w^2(s,t)}{< w^2(s,t) >_t},\tag{7}$$

where $w^2(s,t)$ are the wavelet coefficients evaluated with Equation (4) whilst the symbol $< ... >_t$ indicates time average. Intermittent features were identified by a *LIM* higher than one, while persistent features have a *LIM* lower or equal to this threshold. A series of *LIM* contour maps are presented in Figure 9, for both the baseline and Chevron configurations

reported in the previous sections. At the near-field axial location x/D = 2 presented in
Figure 9a a wide zone dominated by Screech having *LIM* = 1 in the baseline configuration
can be observed. This zone spans from f = 7 kHz up to f = 10 kHz. As expected, this
zone disappears in Figure 9c,e because Chevrons disrupt the feedback loop. Interestingly,
the use of Chevron nozzles also seems to increase the degree of intermittency at the lower
frequencies, by increasing the number of events with *LIM* higher than 1. At downstream
location x/D = 18 *LIM* values results in higher levels due to the jet development, moreover,
the presence of Screech is undetectable for the baseline configuration Figure 9b. In the case
of the Chevron nozzles increased *LIM* value at low frequencies can be observed, this could
be closely connected to a rise in the degree of intermittency and probably related to a more
significant jet development, Figure 9d,f.

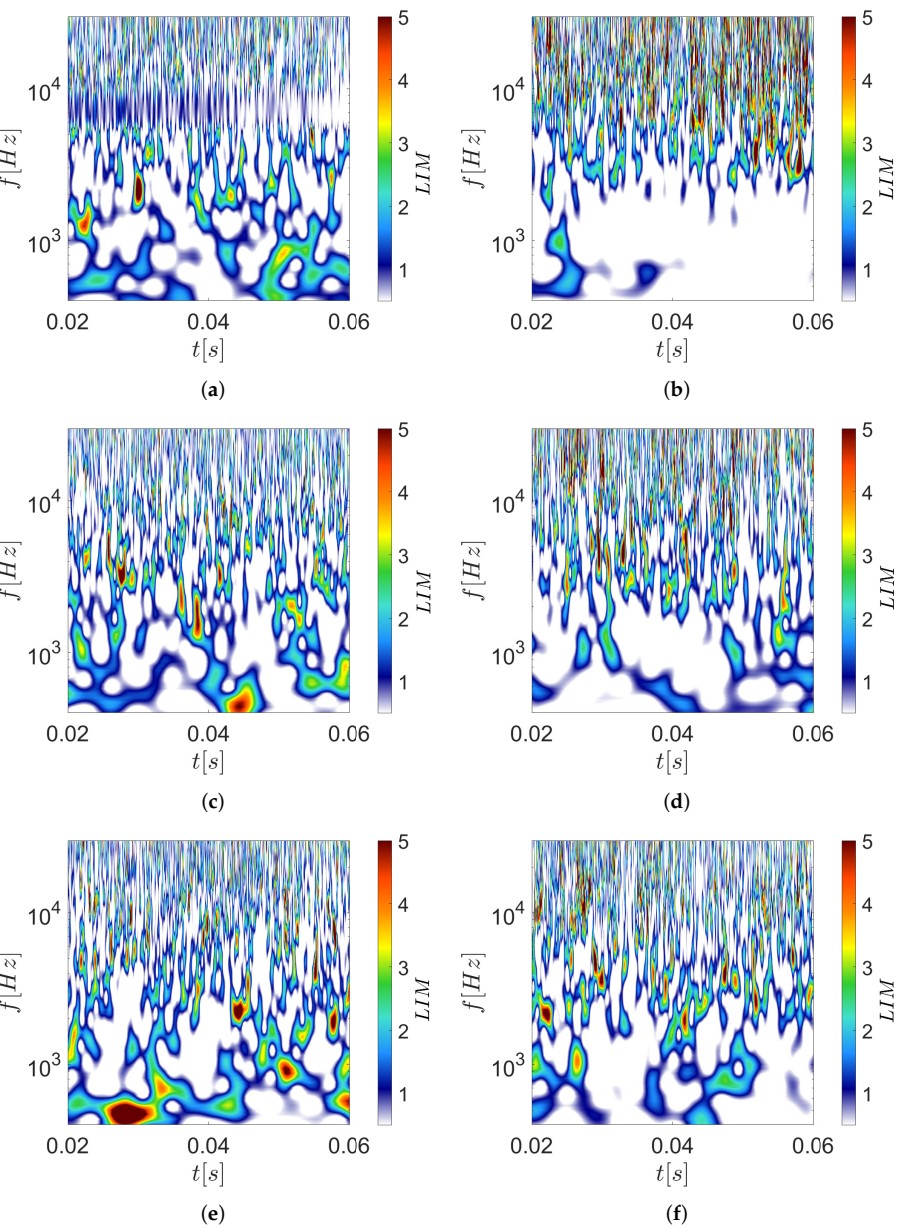

**Figure 9.** *LIM* contour maps at M = 1.3 and h/D = 3.5: (**a**,**b**) Baseline nozzle at x/D = 2 and x/D = 18,
respectively; (**c**,**d**) SMC002 nozzle at x/D = 2 and x/D = 18, respectively; (**e**,**f**) SMC006 nozzle at
x/D = 2 and x/D = 18, respectively.

Following previous studies [4] highly intermittent events could be identified using the square of the *LIM*, named *LIM*2 as reported in the following equation, which is the definition suggested by [28].

$$LIM2(s,t) = \frac{w^4(s,t)}{<w^4(s,t)>_t},$$ 

(8)

*LIM*2 represents a convenient tool to extract those features contributing to the deviation from Gaussianity of the wavelet coefficients and can be interpreted as a time scale-dependent measure of the flatness factor of the signal. Indeed, *LIM*2 is equal to 3 if the PDF is Gaussian, and consequently, the condition *LIM*2 > 3 identifies only those events contributing to the deviation of the PDF from the Gaussian distribution of the wavelet coefficients. Otherwise, if *LIM*2 is lower than 3, it identifies only those events that induce a bi-modal distribution of the wavelet coefficients. This is an efficient tool to highlight the statistical content of a signal when intermittency or bi-modal contributions are confined at specific frequencies.

In the presented case, the Screech feature is characterized in the baseline configuration (see Figure 10a) by *LIM*2, which varies between 1 and 2 with no event above 3. This showed as expected that the Screech signatures has a bi-modal distribution in time, which is generated by the resonant phenomenon of the Screech. However, Chevron nozzles create a series of intermittent events in this frequency region, contributing to the deviation from a Gaussian distribution, increasing the global kurtosis of the signal (see Figure 10c,e). Considering the other frequencies, various effects can be observed: the higher frequency region seems slightly influenced by the Chevrons, while frequency at around $10^3$ increased the number of events with larger *LIM*2 for the Chevron configuration. On the other hand, at x/D = 18, an increase of the *LIM*2 in all the three nozzle configurations (see Figure 10b,d,f) were observed in the higher frequency region while in the lower frequencies, *LIM*2 values decreased, with the larger part of it close to zero. These results are very evident in the baseline configuration. In contrast, some higher *LIM*2 values are detected in both Chevron configurations, which are probably responsible for the higher kurtosis observed in this location (see Figure 2). According to [4], it is important to underline that this behavior was entirely missed by the classical global statistical indicator reported in the first part of the present work.

*3.4. Bi-Variate Wavelet Analysis*

The multivariate wavelet analysis has been performed in the frequency domain using the wavelet coherence evaluated as follows:

$$\gamma_{(}^2 f,t) = \frac{\psi|C_x^*(f,t)C_y(f,t)|^2}{\psi|C_x(f,t)|^2 \cdot \psi|C_y(f,t)|^2}$$ 

(9)

where $|C_x^*(f,t)C_y(f,t)|$ is the wavelet cross-spectrum of two consecutive signals in the axial direction, while $C_x(f,t)$ and $C_y(f,t)$ denote the continuous wavelet transforms of x and y (the two signals) at frequency f and time position t and $\psi$ indicates the smoothing factor. Without the smooth function $\psi$, $\gamma(f,t)$ will be equal to one everywhere. Hence, the wavelet coherence is a normalized scalogram [29,30], which describes the common power of two signals. The advantage in using a cross-wavelet transform with respect to computing a direct coherence is in the locality of the wavelet transform and the different resolutions achievable at the different scales. A series of wavelet coherence contour maps are reported in Figure 11 they are evaluated using signals from consecutive microphones located at different axial locations from the nozzle exit.

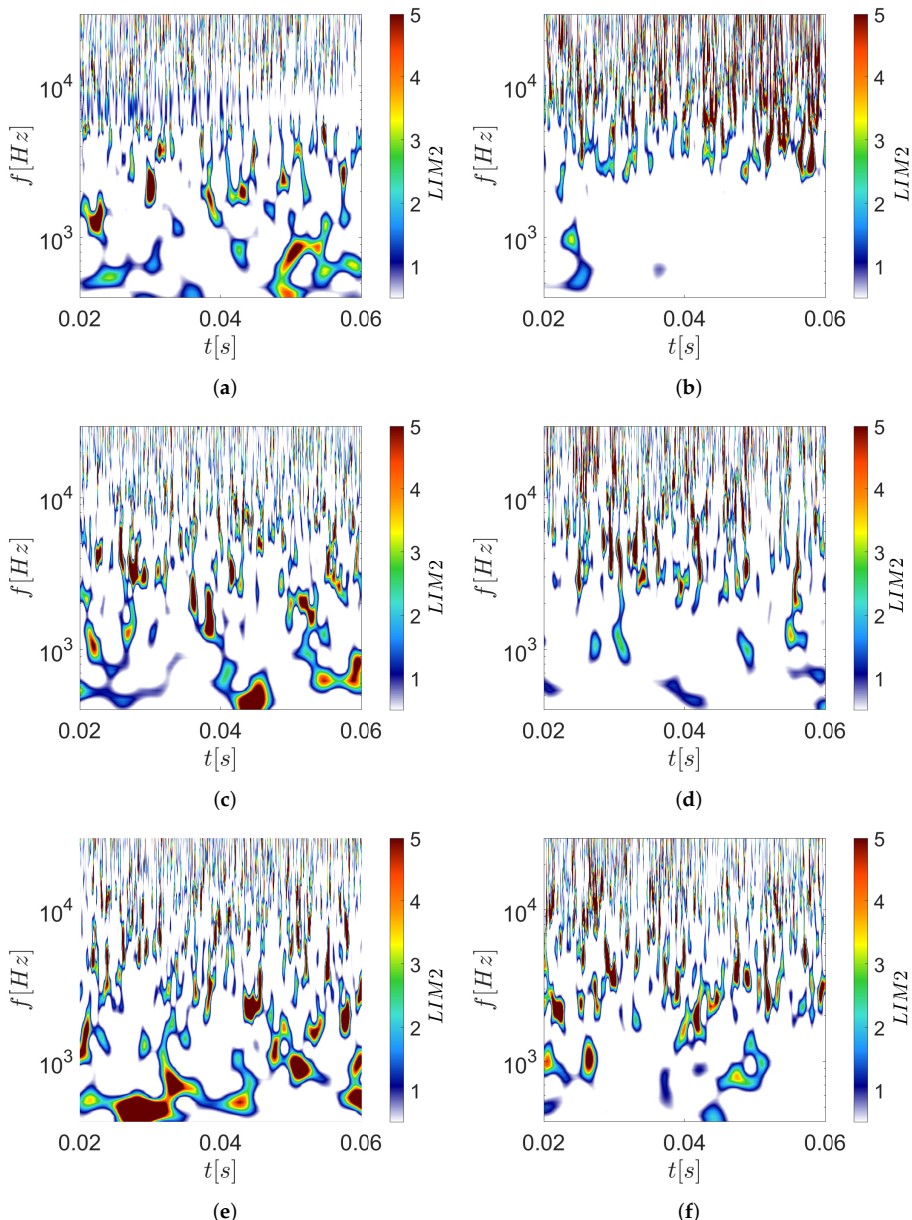

**Figure 10.** *LIM*2 contour maps at M = 1.3 and h/D = 3.5: (**a**,**b**) Baseline nozzle at x/D = 2 and x/D = 18, respectively; (**c**,**d**) SMC002 nozzle at x/D = 2 and x/D = 18, respectively; (**e**,**f**) SMC006 nozzle at x/D = 2 and x/D = 18, respectively.

At first look high levels of coherence can be observed for the baseline configuration, especially at the Screech frequency in Figure 11a when considering two consecutive microphones that are located close to the nozzle exit at x/D = 2–4. The resulting coherence signature is observed to be persistent in time. As expected, when the nozzle configuration is changed to Chevron, the Screech characteristics fade and the coherence value for the above mentioned signature results reduced Figure 11c. These effects remain very similar when the Chevron penetration angle is varied, as shown in Figure 11e. For higher frequencies (>10 kHz) related to the BBSAN, the time step between two consecutive high coherence values results increased with the use of Chevrons. The variation of the Chevrons lobes modifies the low-frequency coherence, reducing it, especially at the mid-frequency range (see Figure 11e,f). This could be ascribed to the variation of the jet velocity gradient, but this needs to be clarified with further investigations with aerodynamic measurements. The analysis has also been repeated for two axial microphones located in the well-developed jet

region (x/D = 16–18). There, Screech signature results are less evident with lower coherence levels (see Figure 11b). High coherence levels in low frequency can be observed in this region due to the large scale structures. For the baseline nozzle, high coherence is found at a frequency as low as 100 Hz. Interestingly, for Chevron nozzles, the coherence levels increase to higher frequencies compared to the baseline configuration. At the downstream location, the effect of the Chevrons results was indistinct on the coherence due to the well-developed jet flow, Figure 11d,f.

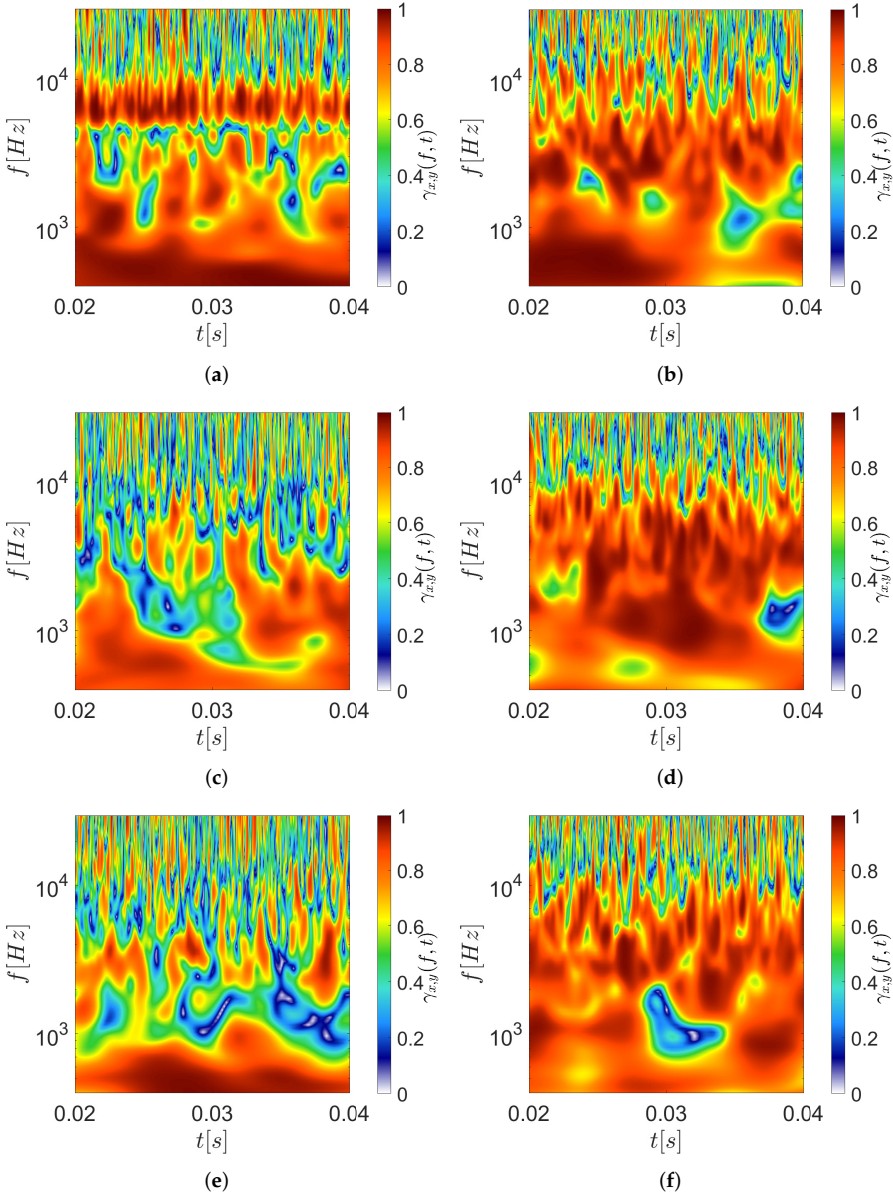

**Figure 11.** Wavelet Coherence contour maps between two consecutive microphones at M = 1.3 and h/D = 3.5: (**a**,**b**) Baseline nozzle having the reference microphone at x/D = 2–4 and x/D = 16–18 respectively; (**c**,**d**) SMC002 nozzle having the reference microphone at x/D = 2–4 and x/D = 16–18, respectively; (**e**,**f**) SMC006 nozzle having the reference microphone at x/D = 2 and x/D = 18, respectively.

The following analysis has been carried out to understand the phase relationship between the two microphones at the frequency related to Screech and BBSAN for the tested configurations. The complex argument can be interpreted as the local relative phase between the two considered pressure signals, in this way, the wavelet cross-spectrum

allows the evaluation of the phase angle between two consecutive signals in both time and frequency domains.

The phase angle $\theta$ has been computed using the following equation:

$$\theta(f,t) = arctan\left(\frac{imag(wcs(f,t))}{real(wcs(f,t))}\right) \tag{10}$$

where $wcs(f,t)$ are the wavelet cross spectrum coefficients.

In contrast to an average phase shift provided by the Fourier analysis, the wavelet phase provides a measure of phase shift in the time domain for each frequency localized feature. Figure 12 shows the phase angle, normalized by $\pi$, between two consecutive microphones in both the time and frequency domain. A phase angle close to $-\pi$ has been detected in the frequency zone dominated by the Screech in the baseline configuration, Figure 12a. This could be due to a persistent upstream event related to the Screech presence with no phase dependency in time. The presence of Chevrons portrays intermittent phase angles as seen in Figure 12c,e. At higher axial locations in the downstream region, the phase angles are characterized by a positive signature at the higher frequencies and a negative signature at the lower frequencies Figure 12b. As for the coherence magnitude at this axial location, the phase angle seems only slightly influenced by the Chevron presence Figure 12d,f. On the other hand, the increasing of the Chevron numbers seems to increase the dominance of the negative phase angle at the lower frequencies, especially close to the nozzle exit. The physical understanding of these results is not trivial and needs further analysis. Regardless of that, in this paper there is the presence of highly intermittent events in both magnitude and phase that should be accounted for in the model to predict both near-field and far-field has been highlighted.

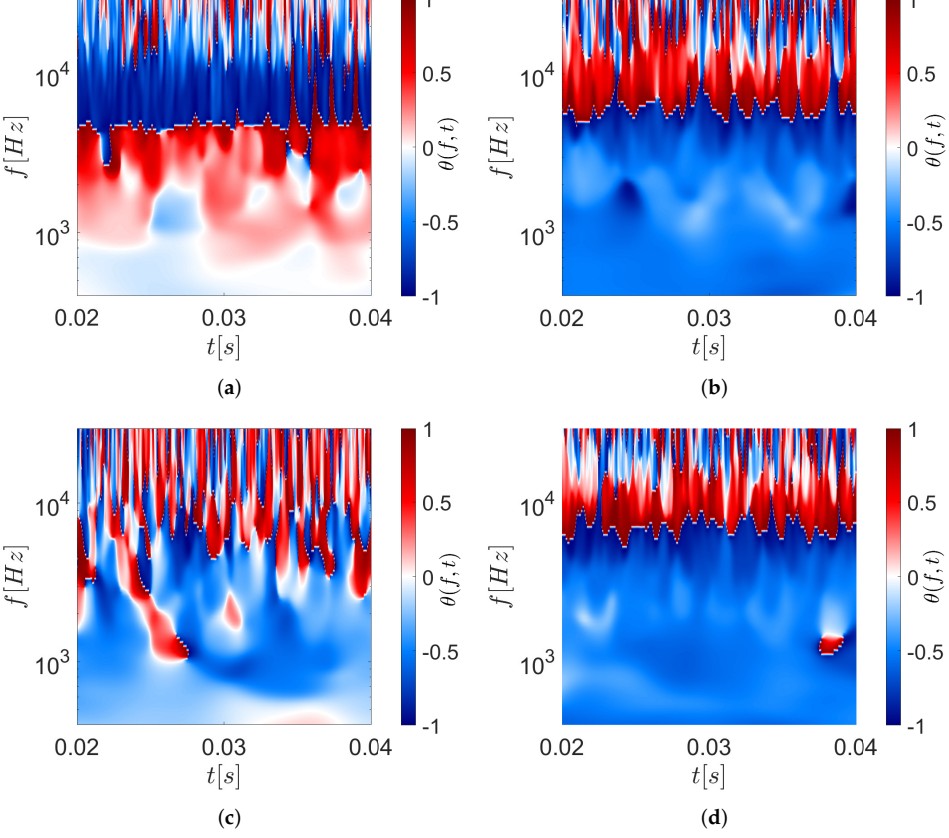

**Figure 12.** *Cont.*

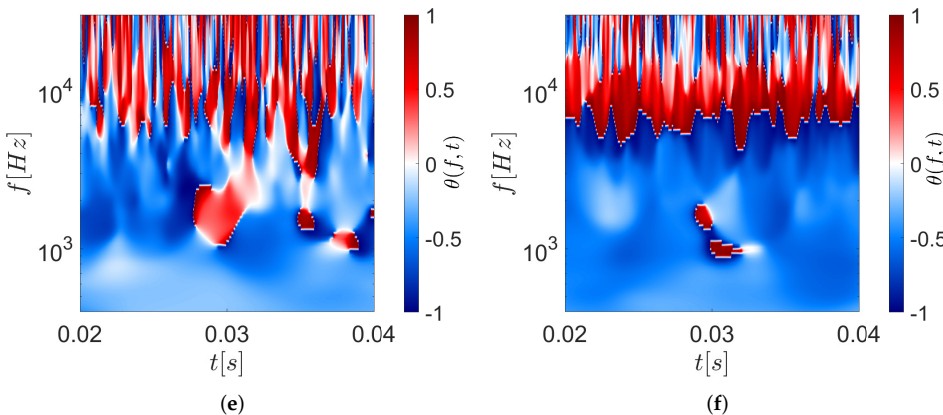

**Figure 12.** Wavelet phase angle contour maps between two consecutive microphones at M = 1.3 and h/D = 3.5: (**a**,**b**) Baseline nozzle having the reference microphone at x/D = 2–4 and x/D = 16–18, respectively; (**c**,**d**) SMC002 nozzle having the reference microphone at x/D = 2–4 and x/D = 16–18, respectively; (**e**,**f**) SMC006 nozzle having the reference microphone at x/D = 2 and x/D = 18, respectively; the reported contours are normalized by $\pi$.

## 4. Conclusions

An in-depth time-frequency analysis using wavelet transform was carried out to deduce and report the stochastic behavior of the jet flow features in the under-expanded supersonic regime with and without Chevrons. Experiments were conducted at the Bristol Jet Aeroacoustic Research Facility at the University of Bristol using a microphone array to attain pressure fluctuations at the vicinity of the jet. Three different nozzle configurations were considered, a baseline case which a classical circular nozzle and two different Chevron nozzles with a different number of Chevron lobes. Data were acquired with the convergent nozzles in under-expanded conditions at M = 1.3.

The conventional spectral level analysis shows the presence of the characteristic Screech tone and its presence in all the presented axial locations. The use of Chevron nozzles eliminated the Screech tone due to the disruption of the feedback loops that drive the tone. The BBSAN was also captured well for the baseline configuration, and it was found to be still persistent in the Chevron configuration. The global statistical analysis highlighted that the Chevron presence modified the statistical distribution of the pressure events close to the nozzle exit by varying the PDF from a bi-modal trend to a Gaussian distribution. Additionally, larger PDF tails were observed at downstream location x/D = 18, ascribed to the increase of the flow development due to the Chevron lobes.

Consecutive analyses were focused on the time evolution of the features responsible for these statistical changes by using the continuous wavelet transform. The resulting wavelet scalogram was dominated by increased wavelet coefficients at the Screech frequency close to the nozzle exit, and the use of Chevrons diminished these signatures. The *LIM* contours underline that this feature is very persistent in time for the round jet and that Chevrons increase the intermittent events at these frequencies. At x/D = 18 from the nozzle exit, the main effect of the Chevrons were the disruption of the low-frequency energetic signatures detected in the baseline configuration. This effect was amplified with increased Chevron lobes, which could be linked to the reduction of the jet mixing noise. An increase in intermittency was also observed in the *LIM* contour maps in this frequency range. These observations were also reconfirmed by the *LIM*2 analysis, which measures the deviation from a Gaussianity distribution of the PDF events related to frequency localized features. Screech was yet again observed to have a bi-modal distribution in time, while the presence of Chevrons disrupts these signatures and generates a series of intermittent events contributing to a deviation from a Gaussian distribution, increasing the signal kurtosis.

Bi-variate wavelet analysis was included to show the local coherence of two consecutive signals in the near-field region within the potential core and in the jet mixing region.

The baseline nozzle showed high levels of time-persistent coherence in the Screech related frequency close to the nozzle exit. The Screech coherence was absent for the Chevron nozzles, other time-frequency signatures were similar for both the tested Chevron nozzles. On the other hand, at the mid-frequency range, the increased number of Chevron lobes reduces the coherence magnitude compared to the round jet. Finally, Screech is less evident in the developed jet region, and the effect of Chevrons results are indistinct. The coherence analysis has also been used to evaluate the local relative phase angle, measuring the detected features' phase shift in both time and frequency domains. The most important findings were the observation of a phase angle close to $-\pi$ at the Screech frequency for the baseline configuration, which became intermittent with the Chevron presence alternating between positive and negative phase angles. No particular effects were detected at higher axial locations.

Overall, an extensive analysis of the intermittent statistics of the supersonic jet noise features has been reported for the first time. A series of efficient tools that can be applied in a more extensive analysis, oriented at providing all the information needed to model the supersonic jet noise sources has been showcased.

**Author Contributions:** Conceptualization, S.M. and H.K.J.; Methodology, S.M.; Experiments H.K.J.; Data Analysis, S.M.; Writing—Original Draft Preparation, S.M. and H.K.J.; Writing—Review & Editing, S.M. and H.K.J. All authors have read and agreed to the published version of the manuscript.

**Funding:** This research received no external funding.

**Data Availability Statement:** The data presented in this study are available on request from the authors.

**Acknowledgments:** Hasan K. J. would like to acknowledge the Engineering and Physical Sciences Research Council (EPSRC) for supporting this research (Grant No. EP/S000917/1). The authors would like to acknowledge Mahdi Azarpeyvand for allocating time for the tests at the aeroacoustic facility in University of Bristol.

**Conflicts of Interest:** The authors declare no conflict of interest.

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
