# Peer review of "A Wavelet-Based Time-Frequency Analysis on the Supersonic Jet Noise Features with Chevrons"

_fluids, doi:10.3390/fluids7030108_

Round 1

Reviewer 1 Report

Current manuscript has conducted a wavelet-based time-frequency analysis on the supersonic jet noise features with chevrons designs. It need major revisions.

1) All the figures are not clear. Need to use the proper format of the curve. And it can clearly show the results for the readers. 

2) In figure 1, the authors need to add the dimensions on the figures;

3) How is the experimental setup? The author need to provide a necessary photo for it;

4) The results shown in Figure 5 to 10 cannot draw any conclusions. The author need to add one more figures to show why chose the three different shapes. and explain which one has the better performance. 

5) Further discussions on the improvements and physical mechanism should be added/ 

Reviewer 2 Report

The authors consider one of topical problem in modern supersonic gas dynamics and aeroacoustics, i.e. decrease of supersonic jet noise, including screech (discrete) tones. Wavelet-based time-frequency analysis is applied for experimental data, results of this study seem novel, and the paper ultimately costs publication. Nevertheless, there are some points of discussion, which should be cleared, and maybe corrected before accepting this paper:

  1. The authors postulated in their Introduction that the supersonic jet is the under-expanded one during take-off, as, for example, at high altitude. It seems strange, because such rocket engine works in not optimal (not correctly expanded) regime during the whole flight. At least, the designers of one-stage rocket engines try to make the jet plume over-expanded (but not strongly, to avoid flow separation) at rocket starting and low-altitude flight.
  2. The authors also postulated in Introduction that shock cell train is inherent to under-expanded jet flows (though over-expanded jets, and even under-expanded jets from the conical nozzles contains shock-cell “barrel” structures).
  3. Authors wrote that screech tones are present in under-expanded jet flows only, but some studies of over-expanded jets [1] revealed that they can present there also, both on moderate and large flow Mach numbers. I recommend taking this fact into account (maybe, to cite the paper [1] as well as fundamental review [2] of supersonic jet screech studies).
  4. I am very bad expert in English, but it seems difficult to read long Conclusion written in one paragraph. Moreover, some usage of capital letters seems voluntary to me, and errors and typos are present in paper text.

References

[1] Gorshkov G.F., Uskov V.N. Self-Sustained Oscillations in Supersonic Overexpanded Impact Jets // Journal of Applied Mechanics and Technical Physics. 2002. Vol. 43. No. 5. Pp.678-682.

[2] Raman G. Supersonic jet screech: Half-century from Powell to the present // Journal of Sound and Vibration. 1999. Vol. 225. No. 3. Pp. 543–571.

Round 2

Reviewer 1 Report

Accepted as it is. 

Reviewer 2 Report

The authors did ncessary amendments, and now their study seems quite satisfactory. I think that the paper can be published in MDPI Fluids journal.